# LARGE BATCH TRAINING OF CONVOLUTIONAL NETWORKS WITH LAYER-WISE ADAPTIVE RATE SCALING

## ABSTRACT

A common way to speed up training of deep convolutional networks is to add computational units. Training is then performed using data-parallel synchronous Stochastic Gradient Descent (SGD) with a mini-batch divided between computational units. With an increase in the number of nodes, the batch size grows. However, training with a large batch often results in lower model accuracy. We argue that the current recipe for large batch training (linear learning rate scaling with warm-up) does not work for many networks, e.g. for Alexnet, Googlenet,... We propose a more general training algorithm based on Layer-wise Adaptive Rate Scaling (LARS). The key idea of LARS is to stabilize training by keeping the magnitude of update proportional to the norm of weights for each layer. This is done through gradient rescaling per layer. Using LARS, we successfully trained AlexNet and ResNet-50 to a batch size of 16K.

## 1 INTRODUCTION

Training of large Convolutional Neural Networks (CNN) takes a lot of time. The brute-force way to speed up CNN training is to add more computational power (e.g. more GPU nodes) and train network using data-parallel Stochastic Gradient Descent, where each worker receives some chunk of global mini-batch (see e.g. Krizhevsky (2014) or Goyal et al. (2017) ). The size of a chunk should be large enough to utilize the computational resources of the worker. So scaling up the number of workers results in the increase of batch size. But using large batch may negatively impact the model accuracy, as was observed in Krizhevsky (2014), Li et al. (2014), Keskar et al. (2016), Hoffer et al. (2017).

Increasing the global batch while keeping the same number of epochs means that you have fewer iterations to update weights. The straight-forward way to compensate for a smaller number of iterations is to do larger steps by increasing the learning rate (LR). For example, Krizhevsky (2014) suggests to linearly scale up LR with batch size. However using a larger LR makes optimization more difficult, and networks may diverge especially during the initial phase. To overcome this difficulty, Goyal et al. (2017) suggested a "learning rate warm-up": training starts with a small LR, which is slowly increased to the target "base" LR. With a LR warm-up and a linear scaling rule, Goyal et al. (2017) successfully trained ResNet-50 [He et al. (2016)] with batch B=8K, see also [Cho et al. (2017)]. Linear scaling of LR with a warm-up is the "state-of-the art" recipe for large batch training.

We tried to apply this linear scaling and warm-up scheme to train AlexNet [Krizhevsky et al. (2012)] on ImageNet [Deng et al. (2009)], but scaling stopped after B=2K since training diverged for large LR-s. For B=4K the accuracy dropped from the baseline 57.6% (B=512) to 53.1%, and for B=8K the accuracy decreased to 44.8%. To enable training with a large LR, we replaced Local Response Normalization layers in AlexNet with Batch Normalization (BN) [Ioffe & Szegedy (2015)]. We will refer to this models AlexNet-BN. BN improved model convergence for large LRs as well as accuracy: for B=8K the accuracy gap decreased from 14% to 2.2%.

To analyze the training stability with large LRs we measured the ratio between the norm of the layer weights and norm of gradients update. We observed that if this ratio is too high, the training becomes unstable. On other hand, if the ratio is too small, then weights don't change fast enough. The layer with largest $\frac{||\nabla W||}{||W||}$ defines the global limit on the learning rate. Since this ratio varies a lot between different layers, we can speed-up training by using a separate LR for each layer. Thus we propose a novel Layer-wise Adaptive Rate Scaling (LARS) algorithm.

There are two notable differences between LARS and other adaptive algorithms such as ADAM (Kingma & Ba (2014)) or RMSProp (Tieleman & Hinton (2012)): first, LARS uses a separate learning rate for each layer and not for each weight, which leads to better stability. And second, the magnitude of the update is defined with respect to the weight norm for better control of training speed. With LARS we trained AlexNet-BN and ResNet-50 with B=16K without accuracy loss.

## 2 BACKGROUND

The training of CNN is done using Stochastic Gradient (SG) based methods. At each step $t$ a mini-batch of $B$ samples $x_i$ is selected from the training set. The gradients of loss function $\nabla L(x_i, w)$ are computed for this subset, and networks weights $w$ are updated based on this stochastic gradient:

$$w_{t+1} = w_t - \lambda \frac{1}{B} \sum\nolimits_{i=1}^{B} \nabla L(x_i, w_t) \tag{1}$$

The computation of SG can be done in parallel by $N$ units, where each unit processes a chunk of the mini-batch with $\frac{B}{N}$ samples. Increasing the mini-batch permits scaling to more nodes without reducing the workload on each unit. However, it was observed that training with a large batch is difficult. To maintain the network accuracy, it is necessary to carefully adjust training hyper-parameters (learning rate, momentum etc).

Krizhevsky (2014) suggested the following rules for training with large batches: when you increase the batch $B$ by $k$ times, you should also increase LR by $k$ times while keeping other hyper-parameters (momentum, weight decay, etc) unchanged. The logic behind **linear LR scaling** is straight-forward: if you increase $B$ by $k$ times while keeping the number of epochs unchanged, you will do $k$ times fewer steps. So it seems natural to increase the step size by $k$ times. For example, let's take $k = 2$. The weight updates for batch size $B$ after 2 iterations would be:

$$w_{t+2} = w_t - \lambda * \frac{1}{B} \left( \sum\nolimits_{i=1}^{B} \nabla L(x_i, w_t) + \sum\nolimits_{j=B+1}^{2B} \nabla L(x_j, w_{t+1}) \right) \tag{2}$$

The weight update for the batch $B_2 = 2 * B$ with learning rate $\lambda_2$:

$$w_{t+1} = w_t - \lambda_2 * \frac{1}{2 * B} \sum\nolimits_{i=1}^{2B} \nabla L(x_i, w_t) \tag{3}$$

will be similar if you take $\lambda_2 = 2 * \lambda$, assuming that $\nabla L(x_j, w_{t+1}) \approx \nabla L(x_j, w_t)$ .

Using the "linear LR scaling" Krizhevsky (2014) trained AlexNet with batch B=1K with minor ($\approx 1\%$) accuracy loss. The scaling of AlexNet above 2K is difficult, since the training diverges for larger LRs. It was observed that linear scaling works much better for networks with Batch Normalization (e.g. Codreanu et al. (2017)). For example Chen et al. (2016) trained the Inception model with batch B=6400, and Li (2017) trained ResNet-152 for B=5K.

The main obstacle for scaling up batch is the instability of training with high LR. Hoffer et al. (2017) tried to use less aggressive "square root scaling" of LR with special form of Batch Normalization ("Ghost Batch Normalization") to train AlexNet with B=8K, but still the accuracy (53.93%) was much worse than baseline 58%. To overcome the instability during initial phase, Goyal et al. (2017) proposed to use **LR warm-up**: training starts with small LR, and then LR is gradually increased to the target. After the warm-up period (usually a few epochs), you switch to the regular LR policy ("multi-steps", polynomial decay etc). Using LR warm-up and linear scaling Goyal et al. (2017) trained ResNet-50 with batch B=8K without loss in accuracy. These recipes constitute the current state-of-the-art for large batch training, and we used them as the starting point of our experiments.

Another problem related to large batch training is so called "generalization gap", observed by Keskar et al. (2016). They came to conclusion that "the lack of generalization ability is due to the fact that large-batch methods tend to converge to sharp minimizers of the training function." They tried a few methods to improve the generalization with data augmentation and warm-starting with small batch, but they did not find a working solution.

## 3  ANALYSIS OF ALEXNET TRAINING WITH LARGE BATCH

We used BVLC[1] AlexNet with batch B=512 as baseline. Model was trained using SGD with momentum 0.9 with initial LR=0.02 and the polynomial (power=2) decay LR policy for 100 epochs. The baseline accuracy is 58.8% (averaged over last 5 epochs). Next we tried to train AlexNet with B=4K by using larger LR. In our experiments we changed the base LR from 0.01 to 0.08, but training diverged with LR > 0.06 even with warm-up [2]. The best accuracy for B=4K is 53.1%, achieved for LR=0.05. For B=8K we couldn't scale-up LR either, and the best accuracy is 44.8% , achieved for LR=0.03 (see Table 1(a) ).

To stabilize the initial training phase we replaced Local Response Normalization layers with Batch Normalization (BN). We will refer to this model as AlexNet-BN. [3]. AlexNet-BN model was trained using SGD with momentum=0.9, weight decay=0.0005 for 128 epochs. We used polynomial (power 2) decay LR policy with base LR=0.02. The baseline accuracy for B=512 is 60.2%. With BN we could use large LR-s even without warm-up. For B=4K the best accuracy 58.9% was achieved for LR=0.18, and for B=8K the best accuracy 58% was achieved for LR=0.3. We also observed that BN significantly widens the range of LRs with good accuracy.

Table 1: AlexNet and AlexNet-BN trained for 100 epcohs: B=4K and B=8K. BatchNorm makes it possible to use larger learning rates, but training with large batch still results in lower accuracy.

<table>
<tr><td colspan="3">(a) AlexNet (warm-up 2.5 epochs)</td><td colspan="3">(b) AlexNet-BN (no warm-up)</td></tr>
<tr><td>Batch</td><td>Base LR</td><td>accuracy,%</td><td>Batch</td><td>Base LR</td><td>accuracy,%</td></tr>
<tr><td>512</td><td>0.02</td><td>58.8</td><td>512</td><td>0.02</td><td>60.2</td></tr>
<tr><td>4096</td><td>0.04</td><td>53.0</td><td>4096</td><td>0.16</td><td>58.1</td></tr>
<tr><td>4096</td><td>0.05</td><td>53.1</td><td>4096</td><td>0.18</td><td>58.9</td></tr>
<tr><td>4096</td><td>0.06</td><td>51.6</td><td>4096</td><td>0.21</td><td>58.5</td></tr>
<tr><td>4096</td><td>0.07</td><td>0.1</td><td>4096</td><td>0.30</td><td>57.1</td></tr>
<tr><td>8192</td><td>0.02</td><td>29.8</td><td>8192</td><td>0.23</td><td>57.6</td></tr>
<tr><td>8192</td><td>0.03</td><td>44.8</td><td>8192</td><td>0.30</td><td>58.0</td></tr>
<tr><td>8192</td><td>0.04</td><td>43.1</td><td>8192</td><td>0.32</td><td>57.7</td></tr>
<tr><td>8192</td><td>0.05</td><td>0.1</td><td>8192</td><td>0.41</td><td>56.5</td></tr>
</table>

Still there is a 2.2% accuracy loss for B=8K. To check if it is related to the "generalization gap" (Keskar et al. (2016)), we looked at the loss gap between training and testing (see Fig. 1). We did not find the significant difference in the loss gap between B=512 and B=8K. We conclude that in this case the accuracy loss was mostly caused by the slow training and was not related to a generalization gap.

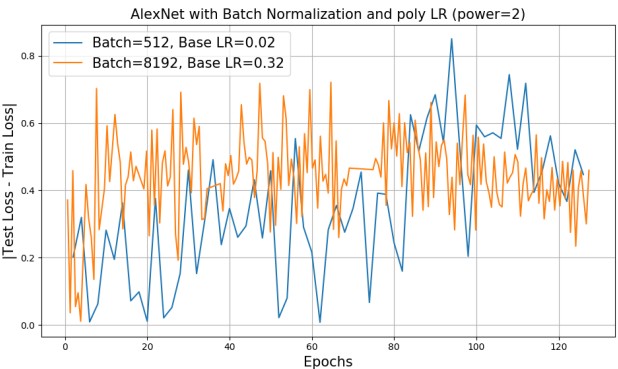

Figure 1: AlexNet-BN: the generalization gap between training and testing loss is practically the same for small (B=256) and large (B=8K) batches.

---

[1]https://github.com/BVLC/caffe/tree/master/models/bvlc_AlexNet

[2]LR starts from 0.001 and is linearly increased it to the target LR during 2.5 epochs

[3] https://github.com/NVIDIA/caffe/tree/caffe-0.16/models/AlexNet_bn

## 4 LAYER-WISE ADAPTIVE RATE SCALING (LARS)

The standard SGD uses the same LR $\lambda$ for all layers: $w_{t+1} = w_t - \lambda \nabla L(w_t)$. When $\lambda$ is large, the update $||\lambda * \nabla L(w_t)||$ can become larger than $||w||$, and this can cause the divergence. This makes the initial phase of training highly sensitive to the weight initialization and to initial LR. We found that the ratio of the L2-norm of weights and gradients $||w||/||\nabla L(w_t)||$ varies significantly between weights and biases, and between different layers. For example, let's take AlexNet after one iteration (Table 2, "*.w" means layer weights, and "*.b" - biases). The ratio $||w||/||\nabla L(w)||$ for the 1st convolutional layer ("conv1.w") is 5.76, and for the last fully connected layer ("fc6.w") - 1345. The ratio is high during the initial phase, and it is rapidly decreasing after few epochs (see Figure 2).

Table 2: AlexNet: The ratio of norm of weights to norm of gradients for different layers at 1st iteration. The maximum learning rate is limited by the layer with smallest ratio.

| Layer | conv1.b | conv1.w | conv2.b | conv2.w | conv3.b | conv3.w | conv4.b | conv4.w |
|---|---|---|---|---|---|---|---|---|
| $||w||$ | 1.86 | 0.098 | 5.546 | 0.16 | 9.40 | 0.196 | 8.15 | 0.196 |
| $||\nabla L(w)||$ | 0.22 | 0.017 | 0.165 | 0.002 | 0.135 | 0.0015 | 0.109 | 0.0013 |
| $\frac{||w||}{||\nabla L(w)||}$ | 8.48 | **5.76** | 33.6 | 83.5 | 69.9 | 127 | 74.6 | 148 |
| Layer | conv5.b | conv5.w | fc6.b | fc6.w | fc7.b | fc7.w | fc8.b | fc8.w |
| $||w||$ | 6.65 | 0.16 | 30.7 | 6.4 | 20.5 | 6.4 | 20.2 | 0.316 |
| $||\nabla L(w)||$ | 0.09 | 0.0002 | 0.26 | 0.005 | 0.30 | 0.013 | 0.22 | 0.016 |
| $\frac{||w||}{||\nabla L(w)||}$ | 73.6 | 69 | 117 | **1345** | 68 | 489 | 93 | 19 |

If LR is large comparing to the ratio for some layer, then training may becomes unstable. The LR "warm-up" attempts to overcome this difficulty by starting from small LR, which can be safely used for all layers, and then slowly increasing it until weights will grow up enough to use larger LRs.

We would like to use different approach. We want to make sure that weights update is small comparing to the norm of weights to stabilize training

$$||\triangle w_t^l|| < \eta * ||w_t^l|| \tag{4}$$

where $\eta < 1$ control the magnitude of update with respect to weights. The coefficient $\eta$ defines how much we "trust" that the value of stochastic gradient $\nabla L(w_t^l)$ is close to true gradient. The $\eta$ depends on the batch size. "Trust" $\eta$ is monotonically increasing with batch size: for example for Alexnet for batch $B = 1K$ the optimal $\eta = 0.0002$, for batch $B = 4K$ - $\eta = 0.005$, and for $B = 8K$ - $\eta = 0.008$. We implemented this idea through defining local LR $\lambda^l$ for each layer $l$:

$$\triangle w_t^l = \gamma * \lambda^l * \nabla L(w_t^l) \tag{5}$$

where $\gamma$ defines a global LR policy (e.g. steps, or exponential decay), and local LR $\lambda^l$ is defined for each layer through "trust" coefficient $\eta < 1$ [4]:

$$\lambda^l = \eta \times \frac{||w^l||}{||\nabla L(w^l)||} \tag{6}$$

Note that now the magnitude of the update for each layer doesn't depend on the magnitude of the gradient anymore, so it helps to partially eliminate vanishing and exploding gradient problems. The network training for SGD with LARS are summarized in the Algorithm 1 [5].

LARS was designed to solve the optimization difficulties, and it does not replace standard regularization methods (weight decay, batch norm, or data augmentation). But we found that with LARS we can use larger weight decay, since LARS automatically controls the norm of layer weights:

$$||w_t|| < ||w_0|| * e^{\eta * \int_0^t \gamma(\tau)d\tau} < ||w_0|| * e^{\frac{\eta * N * S}{B}} \tag{7}$$

where $B$ is min-batch size, $N$ - number of training epochs, and $S$ - number of samples in the training set. Here we assumed that global rate policy starts from 1 and decrease during training over training interval $[0, N * S/B]$.

---

[4] One can consider LARS as a particular case of block-diagonal re-scaling from Lafond et al. (2017).
[5] More details in https://github.com/NVIDIA/caffe/blob/caffe-0.16/src/caffe/solvers/sgd_solver.cpp

---

**Algorithm 1** SGD with LARS. Example with weight decay, momentum and polynomial LR decay.

---

**Parameters:** base LR $\gamma_0$, momentum $m$, weight decay $\beta$, LARS coefficient $\eta$, number of steps $T$
**Init:** $t = 0, v = 0$. Init weight $w_0^l$ for each layer $l$
**while** $t < T$ for each layer $l$ **do**
   $g_t^l \leftarrow \nabla L(w_t^l)$ (obtain a stochastic gradient for the current mini-batch)
   $\gamma_t \leftarrow \gamma_0 * \left(1 - \frac{t}{T}\right)^2$ (compute the global learning rate)
   $\lambda^l \leftarrow \eta * \frac{||w_t^l||}{||g_t^l|| + \beta||w_t^l||}$ (compute the local LR $\lambda^l$)
   $v_{t+1}^l \leftarrow mv_t^l + \gamma_t * \lambda^l * (g_t^l + \beta w_t^l)$ (update the momentum)
   $w_{t+1}^l \leftarrow w_t^l - v_{t+1}^l$ (update the weights)
**end while**

---

## 5 TRAINING WITH LARS

We re-trained AlexNet and AlexNet-BN with LARS for batches up to 32K [6]. To emulate large batches (B=16K and 32K) we used caffe parameter $iter\_size$ [7] to partition batch into smaller chunks. Both Alexnet and Alexnet-BN have been trained for 100 epochs using SGD with momentum=0.9, weight decay=0.0005. We used global learning rate with polynomial decay (p=2) policy and with warm-up (for Alexnet warm-up was 2 epochs, and for Alexnet-BN 5 epochs). We fixed LARS trust $\eta = 0.001$ for all batch sizes, and scaled up initial LR as shown in the Table. 3. But one can instead fix global base LR to 1, and scale up the trust coefficient. For B=8K the accuracy of both networks matched the baseline B=512 (see Figure 2). AlexNet-BN trained with B=16K lost 0.9% in accuracy, and trained with B=32K lost 2.6%.

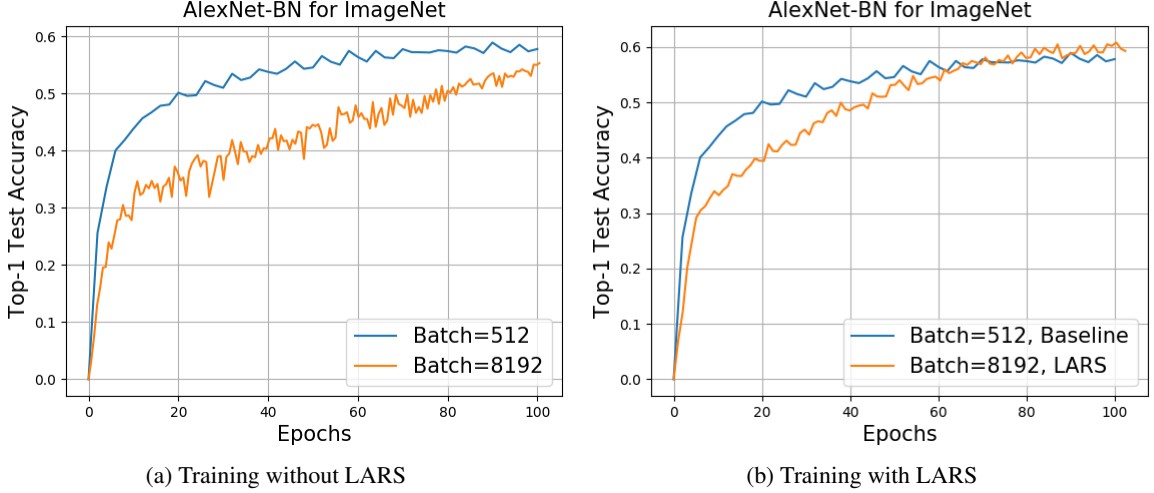

(a) Training without LARS            (b) Training with LARS

Figure 2: Training with LARS: AlexNet-BN with B=8K

We observed that the optimal LR do incresae with batch size, but not in linear or square root proportion as was suggested in theory: there is a relatively wide interval of base LRs which gives the "best" accuracy. For AlexNet-BN with B=16K for example, all LRs from [13;22] interval give almost the same accuracy $\approx 59.3$.

Next we trained ResNet-50, v.1 [He et al. (2016)] with LARS. First we used minimal data augmentation: during training images are scaled to 256x256, and then random 224x224 crop with horizontal flip is taken. All training was done with SGD with momentum 0.9 and weight decay=0.0005 for 100 epochs. We used polynomial decay (power=2) LR policy with LARS and warm-up (5-12 epochs).

---

[6]Training have been done on NVIDIA DGX1 with 8 GPUs.

[7]For example, assume that we want to train model with batch 8K, but only batch 1K fits into GPU memory. In this case we set $iter\_size = 8$, and the weights update is done after gradients for the last chunk are computed.

Table 3: Alexnet and Alexnet-BN training with LARS: the top-1 accuracy as function of batch size

(a) AlexNet with LARS

| Batch | LR | accuracy,% |
|-------|-----|-----------|
| 512 | 2 | 58.7 |
| 4K | 10 | 58.5 |
| 8K | 10 | 58.2 |
| 16K | 14 | 55.0 |
| 32K | 14 | 46.9 |

(b) AlexNet-BN with LARS

| Batch | LR | accuracy,% |
|-------|-----|-----------|
| 512 | 2 | 60.2 |
| 4K | 10 | 60.4 |
| 8K | 14 | 60.1 |
| 16K | 23 | 59.3 |
| 32K | 22 | 57.8 |

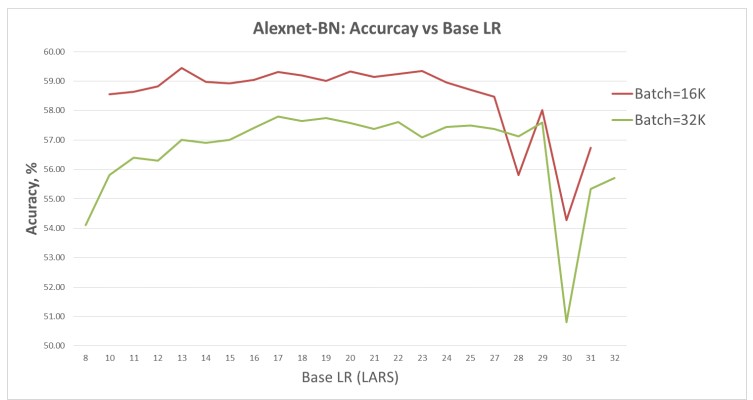

Figure 3: AlexNet-BN, B=16K and 32k: Accuracy as function of LR

During testing we used one model and 1 central crop. The baseline (B=256) accuracy is 73.8% for minimal augmentation. To match the state-of-the art accuracy from Goyal et al. (2017) and Cho et al. (2017) we used the second setup with an extended augmentation with variable image scale and aspect ratio similar to [Szegedy et al. (2015)] . The baseline top-1 accuracy for this setup is 75.4%.

Table 4: ResNet50 with LARS: top-1 accuracy as function of batch size for training with minimal and extended data augmentation

| Batch | $\gamma$ | warm-up | min aug, accuracy,% | max aug, accuracy, % |
|-------|-----|---------|---------------------|----------------------|
| 256 | 4 | N/A | 73.8 | 75.8 |
| 1K | 9 | 5 | 73.3 | 75.4 |
| 8K | 30 | 5 | 73.5 | 75.2 |
| 16K | 33 | 12 | 72.9 | 74.4 |
| 32K | 40 | 12 | 72.5 | 72.5 |

The accuracy with B=16K is 0.7-1.4% less than baseline. This gap is related to smaller number of steps. We will show in the next section that one can recover the accuracy by training for more epochs.

## 6 LARGE BATCH TRAINING: ACCURACY VS NUMBER OF STEPS

When batch becomes large (32K), even models trained with LARS and large LR don't reach the baseline accuracy. One way to recover the lost accuracy is to train longer (see [Hoffer et al. (2017)]). Note that when batch becomes large, the number of iteration decrease. So one way to try to improve the accuracy, would be train longer. For example for Alexnet and Alexnet-BN with B=16K, when we double the number of iterations from 7800 (100 epochs) to 15600 (200 epochs) the accuracy improved by 2-3% (see Table 5). The same effect we observed for Resnet-50: training for additional 100 epochs recovered the top-1 accuracy to the baseline 75.5%.

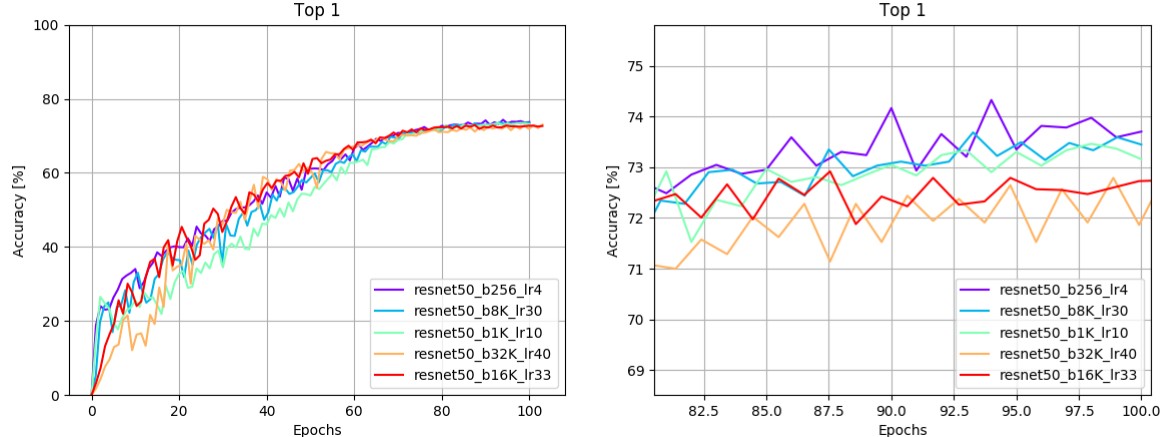

Figure 4: Scaling ResNet-50 (no data augmentation) up to B=32K with LARS.

Table 5: Accuracy vs Training duration

(a) AlexNet, B=16k

| Epochs | accuracy,% |
|--------|-----------|
| 100 | 55.0 |
| 125 | 55.9 |
| 150 | 56.7 |
| 175 | 57.3 |
| 200 | 58.2 |

(b) AlexNet-BN, B=32K

| Epochs | accuracy,% |
|--------|-----------|
| 100 | 57.8 |
| 125 | 59.2 |
| 150 | 59.5 |
| 175 | 59.5 |
| 200 | 59.9 |

(c) ResNet-50, B=16K

| Epochs | accuracy,% |
|--------|-----------|
| 100 | 74.4 |
| 125 | 74.1 |
| 150 | 74.6 |
| 175 | 74.8 |
| 200 | 75.5 |

In general we found that we have to increase the training duration to keep the accuracy. Consider for example Googlenet [Szegedy et al. (2015)]. As a baseline we trained BVLC googlenet [8] with batch=256 for 100 epoch. The top-1 accuracy of this model is 69.2%. Googlenet is deep, so in original paper authors used auxiliary losses to accelerate SGD. We used LARS to solve optimization difficulties so we don't need these auxiliary losses. The original model also has no Batch Normalization, so we used data augmentation for better regularization. The baseline accuracy for B=256 is 70.3% with extended augmentation and LARS. We found that Googlenet is very difficult to train with large batch even with LARS: we needed both large number of epoch and longer ramp-up to scale learning rate up (see Table 6).

Table 6: Googlenet training with LARS

| Batch | $\gamma$ | epochs | warm-up | accuracy,% |
|-------|----------|--------|---------|-----------|
| 256 (no LARS) | 0.02 | 100 | - | 69.2 |
| 256 (LARS) | 4 | 100 | 5 | 70.3 |
| 1K | 6 | 100 | 10 | 69.7 |
| 2K | 7 | 150 | 25 | 71.1 |
| 4K | 10 | 200 | 30 | 69.9 |
| 8K | 10 | 250 | 60 | 69.0 |
| 16K | 10 | 350 | 110 | 67.2 |

## 7 CONCLUSION

Large batch is a key for scaling up training of convolutional networks. The existing approach for large-batch training, based on using large learning rates, leads to divergence, especially during the

---

[8]https://github.com/BVLC/caffe/tree/master/models/bvlc_googlenet

initial phase, even with learning rate warm-up. To solve these difficulties we proposed the new optimization algorithm, which adapts the learning rate for each layer (LARS) proportional to the ratio between the norm of weights and norm of gradients. With LARS the magnitude of the update for each layer doesn't depend on the magnitude of the gradient anymore, so it helps with vanishing and exploding gradients. But even with LARS and warm-up we couldn't increase LR farther for very large batches, and to keep the accuracy we have to increase the number of epochs and use extensive data augmentation to prevent over-fitting.

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
