# OpenReview forum: "Large Batch Training of Convolutional Networks with Layer-wise Adaptive Rate Scaling"
_ICLR.cc/2018/Conference — Reject_

### Official Review · AnonReviewer1 · 2017-11-27
**A layer-wise learning rate is proposed. Some state-of-the-art baselines are missing in comparison.**

**Rating:** 5
**Confidence:** 3

**Review:**

This paper provides an optimization approach for large batch training of CNN with layer-wise adaptive learning rates.
It starts from the observation that the ratio between the L2-norm of parameters and that of gradients on parameters varies
significantly in the optimization,  and then introduce a local learning rate to consider this observation for a more stable and efficient optimization. Experimental results show improvements compared with the state-of-the-art algorithm.

Review:
(1) Pros
The proposed optimization method considers the dynamic self-adjustment of the learning rate in the optimization based on the ratio between the L2-norm of parameters and that of gradients on parameters  when the batch size increases, and shows improvements in experiments compared with previous methods.

(2) Cons
i) LR "warm-up" can mitigate the unstable training in the initial phase and the proposed method is also motivated by the stability but uses a different approach. However, it seems that the authors also combine with LR "warm-up" in your proposed method in the experimental part, e.g., Table 3. So does it mean that the proposed method cannot handle the problem in general?

ii) There is one coefficient that is independent from layers and needs to be set manually in the proposed local learning rate. The authors do not have a detail explanation and experiments about it. In fact, as can be seen in the Algorithm 1, this coefficient can be as an independent hyper-parameter (even is put with the global learning rate together as one fix term).

iii) In the section 6, when increase the training steps, experiments compared with previous methods should be implemented since they can also get better results with more epochs.

iv) Writing should be improved, e.g., the first paragraph in section 6. Some parts are confusing, for example, the authors claim that they use initial LR=0.01, but in Table 1(a) it is 0.02.

---

> ### Author Response · Authors · 2017-12-12
> **Reply to Comment 3**
>
> 1. Comment : "LR "warm-up" can mitigate the unstable training in the initial phase and the proposed method is also motivated by the stability but uses a different approach. However, it seems that the authors also combine with LR "warm-up" in your proposed method in the experimental part, e.g., Table 3. So does it mean that the proposed method cannot handle the problem in general?"
> A: Warm-up alone is not able to mitigate the unstable training for Alexnet. LARS with warm-up can. There is also a new version of algorithm which eliminates warm-up completely
>
> 2. Comment: "There is one coefficient that is independent from layers and needs to be set manually in the proposed local learning rate. The authors do not have a detail explanation and experiments about it. In fact, as can be seen in the Algorithm 1, this coefficient can be as an independent hyper-parameter (even is put with the global learning rate together as one fix term)."
> A: Agree. In the paper we used fixed trust coefficient $eta" and changed learning rate. One can used instead fixed global learning rate policy which does not depend on networks, and scale up only trust coefficient . I will add the explanation to the revised paper.
>
> 3. Comment "In the section 6, when increase the training steps, experiments compared with previous methods should be implemented since they can also get better results with more epochs."
> A: The point of section 6 was to show that there is no "fundamental" limit on the accuracy of large batch training, provided we do train it long enough and regularize well (e.g. increase weight decay or add data augmentation.
>
> 4. Comment 4: "the authors claim that they use initial LR=0.01, but in Table 1(a) it is 0.02"
>     A: typo is fixed in the revised paper.

---

### Official Review · AnonReviewer3 · 2017-11-27
**The paper proposes a heuristic for layer-wise learning rate selection in convolutional networks. The contribution is relatively minor and is not evaluated with sufficient depth.**

**Rating:** 4
**Confidence:** 4

**Review:**

The paper proposes a new approach to determine learning late for convolutional neural networks. It starts from observation that for batch learning with a fixed number of epochs, the accuracy drops when the batch size is too large.  Assuming that the number or epochs and batch size are fixed, the contribution of the paper is a heuristic that assigns different learning late to each layer of a network depending on a ratio of the norms of weights and gradients in a layer.  The experimental results show that the proposed heuristic helps AlexNet and ResNet end up in a larger accuracy on ImageNet data.
  Positives:
- the proposed approach is intuitively justified
- the experimental results are encouraging
  Negatives:
- the methodological contribution is minor
- no attempt is made to theoretically justify the proposed heuristic
- the method introduces one or two new hyperparameters and it is not clear from the experimental results what overhead is this adding to network training
- the experiments are done only on a single data set, which is not sufficient to establish superiority of an approach
  Suggestions:
- consider using different abbreviation (LARS is used for least-angle regression)

---

> ### Author Response · Authors · 2017-12-12
> **Reply to comment 2**
>
> 1. Comment 1: "the methodological contribution is minor"
>     A: We proposed a new training method, which enable training with large batch of networks which is not possible with all other methods (AFAK).
>
> 2. Comment 2: " no attempt is made to theoretically justify the proposed heuristic"
>    A: " Agree, unfortunately most methods used for deep learning don't have formal proof yet"
>
> 3. Comment 3: "the method introduces one or two new hyperparameters and it is not clear from the experimental results what overhead is this adding to network training"
> A: there is one hyper-parameter -  trust coefficient $0<eta<1"$ . I added the explanation how it depends  to revised paper
>
> 4. Comment 4: "the experiments are done only on a single data set, which is not sufficient to establish superiority of an approach"
> A: We focused on Imagenet classification only  for large batch training The results in the paper are for 3 models (Alexnet, Alexnet-BN, and Resnet-50).  I will add Googlenet results for completeness.
>
> 5.  Suggestions: " consider using different abbreviation (LARS is used for least-angle regression) "
>  A: Agree, probably LARC (Layer-wise Adaptive Rate Control" would be better, but the algorithm was already implemented in nvcaffe when we realized that there is a name collision.

---

### Official Review · AnonReviewer2 · 2017-11-28
**This paper proposes a training algorithm based on Layer-wise Adaptive Rate Scaling (LARS) to overcome the optimization difficulties for training with large batch size. The authors use a linear scaling and warm-up scheme to train AlexNet on ImageNet. The results show promising performance when using a relatively large batch size. The presented method is interesting. However, the experiments are poorly organized since some necessary descriptions and discussions are missing.**

**Rating:** 5
**Confidence:** 5

**Review:**

This paper proposes a training algorithm based on Layer-wise Adaptive Rate Scaling (LARS) to overcome the optimization difficulties for training with large batch size. The authors use a linear scaling and warm-up scheme to train AlexNet on ImageNet. The results show promising performance when using a relatively large batch size. The presented method is interesting. However, the experiments are poorly organized since some necessary descriptions and discussions are missing. My detailed comments are as follows.

Contributions：

1.	The authors propose a training algorithm based LARS with the adaptive learning rate for each layer, and train the AlexNet and ResNet-50 to a batch size of 16K.
2.	The training method shows stable performance and helps to avoid gradient vanishing or exploding.

Weak points:

The training algorithm does not overcome the optimization difficulties when the batch size becomes larger (e.g. 32K), where the training becomes unstable, and the training based on LARS and warm-up can’t improve the accuracy compared to the baselines.

Specific comments:

1.	In Algorithm 1, how to choose $ \eta $ and $ \beta $ in the experiment?
2.	Under the line of Equation (3), $ \nabla L(x_j, w_{t+1}) \approx L(x_j, w_{t}) $ should be $ \nabla L(x_j, w_{t+1}) \approx \nabla L(x_j, w_{t}) $.
3.	How can the training algorithm based on LARS improve the generalization for the large batch?
4.	In the experiments, what is the parameter iter_size? How to choose it?
5.	In the experiments, no descriptions and discussions are given for Table 3, Figure 4, Table 4, Figure 5, Table 5 and Table 6. The authors should give more discussions on these tables and figures. Furthermore, the captions of these tables and figures confusing.
6.	On page 4, there is a statement “The ratio is high during the initial phase, and it is rapidly decreasing after few epochs (see Figure 2).” This is quite confusing, since Figure 2 is showing the change of learning rates w.r.t. training epochs.

---

> ### Author Response · Authors · 2017-12-11
> **Reply to Comment 1**
>
> Q: "Weak points: The training algorithm does not overcome the optimization difficulties when the batch size becomes larger (e.g. 32K), where the training becomes unstable, and the training based on LARS and warm-up can’t improve the accuracy compared to the baselines"
> A:
> 1) Standard recipe "increase learning rate proportionally to batch size" does not work  for such networks as Alexnet and Googlenet even with warm-up. LARS is the only algorithm (AFAK) that allows to train Alexnet with batch > 2K to the same accuracy as for small batches
> 2) We added Appendix with data on  LARS performanse comparing to other "Large Batch training" methods for Resnet-50.
>
> Specific comments:
> 1) Q: "In Algorithm 1, how to choose $ \eta $ and $ \beta $ in the experiment?"
> A: $ $0<\eta<1 $, and it depends on the batch size $B$. It grows with B: for example for Alexnet with B=1K  the optimal $\eta=0.002$, with B=4K  the optimal $\eta=0.005$, with B=4K  the optimal $\eta=0.008$,...  Weight decay $\beta$ is chosen as usual. We found that with large batch it's beneficial to increase weight decay to improve the regularization
> 2) typo: fixed in the revised paper
> 3) Q: "How can the training algorithm based on LARS improve the generalization for the large batch?"
> A: LARS does not replace standard regularization methods (weight decay, batch norm, or data augmentation).  But we found that with LARS we can use larger weight decay than usual,  since LARS automatically limits the norm of weights during training: $|| W(T)|| <= ||W(0)|| * exp \int_{0}^{T} \gamma(t) dt$.
> 4) Q: In the experiments, what is the parameter iter_size? How to choose it?
>     A: iter_size is used in caffe to emulate large batch if batch does not fit into GPU DRAM.  For example if the batch which fits in GPU memory is 1K, and we want to use B=8K, then iter_size=8.
> 5) and 6) We will add more explanation to the revised paper

---

### Decision · Program_Chairs · 2018-01-29
**ICLR 2018 Conference Acceptance Decision**

**Decision:**

Reject

**Comment:**

Pros:
+ The proposed large-batch, synchronous SGD method is able to generalize at larger batch sizes than previous approaches (e.g., Goyal et al., 2017).

Cons:
- Evaluation on more than one task would make the paper more convincing.
- The addition of more hyperparameters makes the proposed algorithm less appealing.
- Some theoretical justifiction of the layer-wise rate scaling would help.
- It isn't clear that the comparison to Goyal et al., 2017 is entirely fair, because that paper also had recommendations for the implementation of batch normalization, weight decay, and a momentum correction as the learning rate is scaled up, but this submission does not address any of those.

Although the revised paper addressed many of the reviewers' concerns, they still did not feel it was quite strong enough to be accepted to ICLR.